# Persistent spatial clustering and predictors of pediatric La Crosse virus neuroinvasive disease risk in eastern Tennessee and western North Carolina, 2003–2020

Corey A. Day [1¤]*, Agricola O. Odoi[2], Abelardo Moncayo[3], Michael S. Doyle[4], Carl J. Williams[4], Brian D. Byrd[5], Rebecca T. Trout Fryxell[1]

1 Entomology and Plant Pathology, University of Tennessee, Knoxville, Tennessee, United States of America, 2 Department of Biomedical and Diagnostic Sciences, University of Tennessee, Knoxville, Tennessee, United States of America, 3 Tennessee Department of Health, Nashville, Tennessee, United States of America, 4 North Carolina Division of Public Health, Raleigh, North Carolina, United States of America, 5 Environmental Health Sciences, Western Carolina University, Cullowhee, North Carolina, United States of America

¤ Current address: Entomology & Plant Pathology Department, University of Tennessee, Knoxville, Tennessee, United States of America

* cday11@vols.utk.edu

**Data Availability Statement:** De-identified, aggregated data and associated statistical code for reproducing the purely spatial cluster analysis and

## Abstract

The combined region of eastern Tennessee and western North Carolina has a persistently high risk of pediatric La Crosse virus neuroinvasive disease (LACV-ND). To guide public health intervention in this region, the objectives of this retrospective ecological study were to investigate the geographic clustering and predictors of pediatric LACV-ND risk at the ZIP code tabulation area (ZCTA) level. Data on pediatric cases of LACV-ND reported between 2003 and 2020 were obtained from Tennessee Department of Health and North Carolina Department of Health and Human Services. Purely spatial and space-time scan statistics were used to identify ZCTA-level clusters of confirmed and probable pediatric LACV-ND cases from 2003–2020, and a combination of global and local (i.e., geographically weighted) negative binomial regression models were used to investigate potential predictors of disease risk from 2015–2020. The cluster investigation revealed spatially persistent high-risk and low-risk clusters of LACV-ND, with most cases consistently reported from a few high-risk clusters throughout the entire study period. Temperature and precipitation had positive but antagonistic associations with disease risk from 2015–2020, but the strength of those relationships varied substantially across the study area. Because LACV-ND risk clustering in this region is focally persistent, retroactive case surveillance can be used to guide the implementation of targeted public health intervention to reduce the disease burden in high-risk areas. Additional research on the role of climate in LACV transmission is warranted to support the development of predictive transmission models to guide proactive public health interventions.

predictor investigation are available at https://doi.org/10.5281/zenodo.10999145.

**Funding:** This work was supported by the University of Tennessee Department of Entomology and Plant Pathology to CAD. BDB, MSD and CJW were supported in part by the Epidemiology and Laboratory Capacity for Prevention and Control of Emerging Infectious Diseases (ELC) funding (NU50CK000530: Centers for Disease Control and Prevention) awarded to the North Carolina Department of Health and Human Services, and AM was supported in part by ELC funding (NU50CK000528: Centers for Disease Control and Prevention) awarded to the Tennessee Department of Health. The funders had no role in study design, data collection and analysis, decision to publish, or preparation of the manuscript.

**Competing interests:** The authors have declared that no competing interests exist.

## Author summary

La Crosse virus (LACV) is the most common cause of mosquito-borne neuroinvasive disease (e.g., encephalitis or meningitis) among children in the United States, but the spatial patterns and population-level risk factors for LACV infections are not well understood. In this study, we investigated the spatial patterns and areal risk factors of pediatric LACV neuroinvasive disease (LACV-ND) in eastern Tennessee and western North Carolina, a persistent hotspot of LACV-ND. We found that from 2003–2020, the highest risks of pediatric LACV-ND consistently occurred in a few areas within the study region. We also found that precipitation and temperature had positive but antagonistic relationships with area-level risk from 2015–2020, although the strength of those associations varied significantly across the study area. The results of this study have important implications for public health policy. Most notably, because LACV-ND consistently occurs in the same geographic areas over extended time periods, areas where cases have been previously reported should be prioritized for disease prevention programs to prevent continuous transmission. This study also provides initial evidence that climate may be related to the distribution of LACV-ND risk, but additional research is needed to better understand the complex relationship between climate and LACV epidemiology.

## Introduction

La Crosse virus (LACV) is the leading cause of arboviral neuroinvasive disease among children in the United States (US) [1]. The virus is primarily maintained in an enzootic cycle by the eastern tree hole mosquito, *Aedes triseriatus* (Say, 1823), with accessory transmission from invasive *Ae. albopictus* (Skuse, 1895) and *Ae. japonicus* (Theobald, 1901) through a combination of horizontal (e.g., mosquito-mammal-mosquito, or male-female mosquito via mating) and vertical (i.e., mother-to-offspring) transmission mechanisms [2–4]. Human infections with LACV are usually asymptomatic, but severe infections can cause neuroinvasive diseases including encephalitis, meningitis, meningoencephalitis, and acute flaccid paralysis [5–7]. Approximately 90% of reported La Crosse virus neuroinvasive disease (LACV-ND) occurs in children under the age of 18 years, and although it is rarely fatal, pediatric LACV-ND often results in long-term neurological sequelae such as recurring seizures and reduced intelligence quotient (IQ) that can substantially reduce the affected child's quality of life [7–9]. In addition to its health consequences, LACV-ND can also have a substantial economic toll, with the combined costs of acute treatment and management of long-term sequelae capable of exceeding $5 million for a single case (2023 inflation-adjusted USD) [8].

The geographic focus of reported LACV-ND within the eastern US has changed substantially in recent decades [9–13]. Most cases were reported in the upper Midwest and Ohio until the 1990s, when LACV-ND emerged in the socioeconomically vulnerable southern Appalachian region [9,11,12,14]. Since then, geographically clustered counties in Ohio, West Virginia, and the combined region of eastern Tennessee (TN) and western North Carolina (NC) have reported most LACV–ND cases, while risk in the upper Midwest has diminished [9–11]. The cause of this geographic shift is unclear; the establishment of invasive vectors is a leading hypothesis, but it is also likely that the emergence of LACV-ND in Appalachia was a product of increased regional awareness rather than a true range expansion of LACV, as low rates of LACV-ND had been reported in NC as early as 1964 [3,12,15–17]. The environmental factors which limit the distribution of LACV-ND are also poorly understood, as the geographic ranges of LACV vectors and wildlife hosts exceed the typical distribution of reported LACV-ND [18–22].

There are no clinical methods for preventing LACV-ND, so reducing human exposure to the bites of LACV-infected mosquitoes is the best available intervention to prevent disease. Unfortunately, a lack of infrastructure for vector surveillance and control throughout Appalachia has allowed LACV-ND to persist in high-risk areas with minimal prevention efforts [15]. A potential approach to reducing LACV-ND with limited resources is implementing highly targeted public health interventions, such as vector control and community education, in areas where LACV-ND has been identified. There is evidence that LACV-ND cases tend to be spatially aggregated across extended time periods, indicating that reactive interventions may be appropriate several years after LACV transmission is documented at a residence [23,24]. However, previous studies of LACV-ND spatial patterns have lacked either a sufficient temporal range [10,25,26] or spatial resolution [11] to determine the spatial-temporal persistence of elevated disease risk, so additional research is needed to determine the appropriate period and scale of reactive interventions.

A better understanding of the area-level factors that are associated with the distribution of LACV-ND would also be useful for guiding targeted disease prevention efforts. A few studies have investigated population-level (i.e., county level or tract level) demographic and socioeconomic risk predictors, but no studies have investigated the environmental determinants of areal LACV-ND risk [11,14,27,28]. The environmental risk factors of LACV-ND have been investigated more extensively at the household level, where the presence of natural and artificial habitats for immature mosquitoes, proximity to forests, and abundance of LACV vectors are all associated with LACV-ND risk [29–33]. However, these variables are not easily generalized to broader spatial scales, and there remains a need for research that aims to identify population-level environmental risk factors to guide predictive models and targeted interventions.

This study aimed to address the need for a robust investigation of the spatial patterns and predictors of pediatric LACV-ND risk within the combined region of eastern TN and western NC, a persistent high-risk area for LACV-ND in Appalachia [11]. This region has maintained a disproportionate burden of reported cases since the emergence of LACV-ND in southern Appalachia, with select counties routinely falling within high-risk spatial clusters [3,10,11], but little is known about long-term patterns of geographic clustering or the population-level risk factors of pediatric LACV-ND at spatial scales finer than the county level. To address those knowledge gaps, the objectives of this study were to [1] investigate the spatial-temporal clustering of pediatric LACV-ND risk from 2003–2020 and [2] explore potential environmental, socioeconomic, and demographic predictors of risk from 2015–2020. The results of this study will be useful for guiding LACV-ND prevention and identifying key drivers of spatially associated risk in eastern TN and western NC.

## Methods

### Ethics statement

This study was first approved by the Tennessee Department of Health Institutional Review Board (TDH IRB 2021–0314). After initial approval by the TDH IRB, the University of Tennessee, Knoxville Institutional Review Board approved the inclusion of North Carolina data (UTK IRB-22-07079-XP). The North Carolina Department of Health and Human Services relied on the UTK IRB approval as part of an overriding data use agreement. Participant consent was not obtained because this research did not involve greater than minimal risk to participants' privacy, rights, or welfare.

### Study area, case definition, and case data

This study was performed at the ZIP code tabulation area (ZCTA) level in the combined study area of eastern TN and western NC. The study area includes 59 counties, all within the

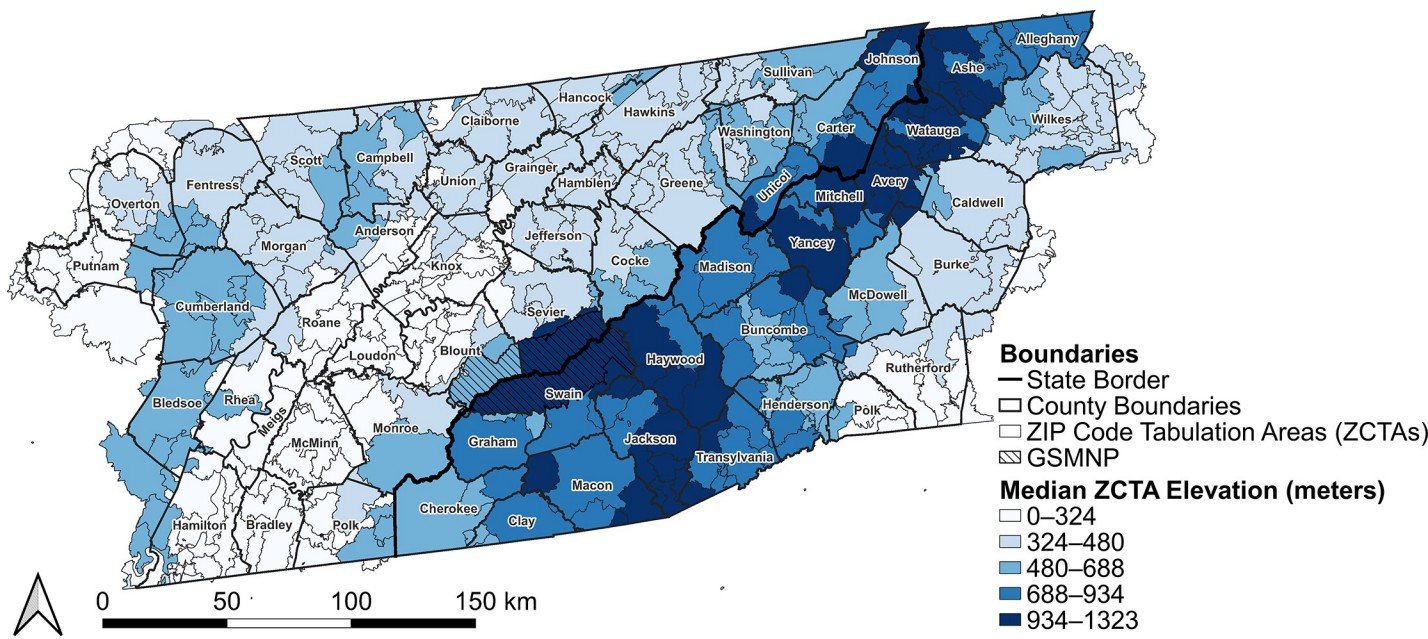

**Fig 1.** Study area of eastern Tennessee (left of state border) and western North Carolina (right of state border) with county names, county boundaries, 2010 ZIP Code Tabulation Area (ZCTA) boundaries, ZCTA-level median elevation, and visual indicator of the Great Smoky Mountains National Park (GSMNP). Base layers were derived from the 2010 United States Census Bureau TIGER/Line files [38].

Appalachian region, and includes the Great Smoky Mountains National Park (GSMNP). All ZCTAs that intersected with any of the 59 counties were included in the study area (Fig 1). As of 2020, the study area had a population of approximately 2,733,474 people with an overall population density of 42.3 people per km$^2$ [34]. The elevation in the study area ranged from 141.6 meters to 2,036.74 meters based on 1m digital elevation maps obtained from the United States Geological Survey LiDAR Explorer Map [35].

This study investigated probable and confirmed cases of LACV-ND. Arboviral infections are considered neuroinvasive when the patient presents with encephalitis, meningitis, acute flaccid paralysis, or other disfunctions of the central or peripheral neurologic system [36]. Probable LACV-ND cases are reported when patients present with neuroinvasive disease and LACV-specific IgM antibodies are identified in cerebrospinal fluid or serum, but no other virus-specific tests are conducted for additional arboviruses that are endemic in the exposure area (e.g., West Nile virus). Confirmed cases must meet additional laboratory criteria including isolation of LACV or demonstration of LACV-specific antigen in nucleic acid or tissue, a 4-fold change in LACV-specific antibody titers in paired sera, or identification of LACV IgM antibodies followed by confirmatory LACV-specific neutralizing antibodies in cerebrospinal fluid or serum [36]. The Tennessee Department of Health and North Carolina Department of Health and Human Services provided data on probable and confirmed cases of LACV-ND reported in the states of Tennessee and North Carolina from 2003 to 2020. The data included the date of symptom onset, age, sex, and home address at time of diagnosis. Cases were aggregated by ZIP code for geographical analyses.

## Objective 1: Cluster investigation

**Data preparation.** Demographic data and cartographic boundaries were obtained from US Census Bureau surveys using the National Historical Geographic Information Service data

finder [37,38] (S1 Table). The ZCTA-level populations 18 years and under (i.e., under-19 population) from the 2000, 2010, and 2020 decennial census surveys, standardized to 2010 ZCTAs to account for changes in ZCTA boundaries over time [39], were used as denominators to compute cumulative incidence risks (CI) and as underlying populations in cluster analyses. Cartographic base layers were derived from the 2010 US Census Bureau TIGER/LINE files [38].

**Descriptive analyses.** The total number of reported cases in eastern TN and western NC were calculated by age groups to analyze the age distribution of LACV-ND in the study area. Stacked bar charts were used to visualize the annual number of cases and the percentage of total cases reported per month in eastern TN and western NC to analyze overall temporal trends and assess differences between the states.

The raw CI of LACV-ND per 10,000 people aged 18 years and younger was calculated for the overall study area and ZCTAs for the periods of 2003–2011 (using the 2000 decennial census under-19 population) and 2012–2020 (using the 2010 decennial census under-19 population). The raw ZCTA-level CIs from 2003–2011 and 2012–2020 were visualized as choropleth maps using Jenks classification scheme to assess changes in the incidence and spatial distribution of cases throughout the study period [40]. To better visualize the distribution of risk in areas with small underlying populations, smoothed CIs were calculated for both periods using spatial empirical Bayes smoothing with the R package 'spdep' [41,42]. The smoothed risks were also visualized as choropleth maps using Jenks classification scheme [40].

**Purely spatial clusters.** Tango's restricted flexibly shaped scan statistic was implemented with the R package 'rflexscan' version 1.0.0 in R version 4.0.3 to identify high-risk purely spatial clusters of pediatric LACV-ND [43–46]. The restricted flexible scan statistic identifies areas of arbitrary shapes that have significantly higher risk than the study area overall. Purely spatial clusters were investigated for two time periods, 2003–2011 and 2012–2020, to compare the distribution of clusters throughout the entire study period. Separate Poisson models were fitted for each time period using the total cases from 2003–2011 or the total cases from 2012–2020 [43,44]. The under-19 populations from the 2000 and 2010 decennial census surveys were specified as the background populations for the 2003–2011 and 2012–2020 models, respectively. Both models used a restricted log-likelihood ratio and 999 Monte Carlo replications to test for clusters, with the maximum cluster size restricted to 30 ZCTAs [47]. Significant clusters ($p < 0.05$) were displayed as choropleth maps for both time periods, with colors denoting cluster rankings based on likelihood ratio test statistics [44,45]. To better visualize changes in the distribution of high-risk clusters over time, an additional choropleth map was used to visualize the ZCTAs that were in a cluster during one or both time periods.

**Space-time clusters.** Kulldorff's space-time scan statistic, implemented with SaTScan version 10.1.2, was used to identify space-time clusters of pediatric LACV-ND across the entire study period (2003–2020) [48,49]. The space-time scan statistic uses a moving cylindrical window method to identify clusters, where the circular base is the geographic size of the scanning window and the height of the cylinder is the temporal scanning window. The number of events within the scanning window is compared to the number expected based on the observed number of events outside the scanning window [48,49]. A significant space-time cluster is one where CI within a specific area and period (i.e., cylindrical window) is significantly greater than expected based on the CI outside that window [50]. Separate space-time cluster analyses were conducted to identify high-risk and low-risk clusters to allow different maximum cluster sizes for each analysis. For the high-risk cluster analysis, the maximum cluster size was specified to include no more than 5% of the total population in the study area to identify the most concentrated clusters to direct potential public health response [50]. For the low-risk cluster analysis, the maximum cluster size included 20% of the population to avoid the identification

of many small, adjacent clusters in large regions with excessively low risk [50]. The maximum temporal window was specified as 90% of the study period (16 years) in both analyses to facilitate detection of long-term persistent clusters; 90% of the total period is the maximum allowed in SaTScan for space-time clusters [50]. The ZCTA-level under-19 population for the years 2000, 2010, and 2020 were included in the analyses to allow the underlying population to vary across the temporal window [50]. High-risk and low-risk clusters with p < 0.05 were visualized on a single choropleth map.

### Objective 2: Predictor investigation

**Selection of potential predictors.** Potential predictors of LACV-ND CI were investigated for probable and confirmed cases of pediatric LACV-ND from 2015 to 2020 to explore ZCTA-level risk factors for recent cases. Potential predictor variables were selected based on existing biological knowledge and evidence of associations with LACV-ND risk in previous studies (S2 Table). The mean, standard error of the mean, median, and the interquartile range (IQR) was calculated for all potential predictor variables for the entire study area.

The predictor investigation included the population aged 19 years and younger (i.e., under-20 population) because the 2020 5-year American Community Survey provides population estimates in five-year age groups (e.g., 15 to 19 years) and does not include an estimate for the under-19 population. To account for this, the definition of pediatric LACV-ND was expanded to include 19-year-olds for the predictor investigation; this decision resulted in the addition of one case aged 19 years.

**Data preparation.** All socioeconomic and demographic predictors were obtained from the 2020 5-year American Community Survey. The sociodemographic predictors considered for LACV-ND risk were percentage of the population living below the poverty line, median age of housing, population density, percentage of the under-20 population that is male, and percentage of vacant houses [11,27,51,52] (see S2 Table for rationale supporting each variable). Cartographic base layers were derived from the 2020 United States Census Bureau TIGER/LINE files [53].

The following land cover variables were assessed for associations with LACV-ND: percent of developed land in 2019, percentage of forested land in 2019, and change in percentage developed land from 2001 to 2019 [16,32,54–56] (S2 Table). Gridded land cover data were obtained at a 30m resolution from the National Land Cover Database (NLCD) [57]. Forested land included all NLCD forest categories (deciduous forest, evergreen forest, and mixed forest), and developed land included the low intensity, medium intensity, and high intensity developed land categories [58]. The land cover data were clipped to the study area using QGIS version 3.24.3 using the Clip Raster function, and the Zonal Histogram tool was used to calculate the percentage of each ZCTA containing forested land in 2019 [59]. The percentage change in developed land from 2001 to 2019 was calculated by subtracting the percentage developed land in 2001 from the developed land in 2019 for each ZCTA.

The climate variables included mean cumulative precipitation, temperature, and dew point in August from 2015–2020 [58–63] (S2 Table). August was chosen because it is typically the peak month for reported LACV-ND [9]. Average temperature, cumulative precipitation, and dew point in August for the years 2015–2020 were obtained from the PRISM Climate Group (PRISM Climate Group, Oregon State University, http://prism.oregonstate.edu, accessed January 2024) at a 4km resolution. The Cell Statistics tool in QGIS was used to create aggregated rasters for each climate variable based on the mean cell value across each year (e.g., mean precipitation in August from 2015–2020), and the Zonal Statistics tool was used to calculate ZCTA-level means from the aggregated rasters for each variable [59]. Median elevation was also investigated to control for

potential climatic effects that were not captured by the temperature and precipitation variables (S2 Table). Digital elevation maps for the study area were obtained from the United States Geological Survey LiDAR Explorer Map at a 1m resolution [35]. The Zonal Statistics tool in QGIS was used to calculate the median elevation for each 2020 ZCTA.

**Global model.** A global model, i.e., a model with one set of coefficients for the entire study area, was used to identify significant predictors of ZCTA-level LACV-ND CI from 2015–2020. Prior to model fitting, correlation analyses were conducted on all pairs of potential predictor variables, and only one variable was retained from any pair of predictors with r > | 0.70|. Unconditional bivariate associations between predictors and LACV-ND CI were assessed by fitting univariable negative binomial models with a log-link function for each remaining predictor [64]. In each model, the dependent variable was the number of cases in the under-20 population from 2015–2020, specifying the natural log of the total under-20 population as the offset to effectively analyze CI. Any predictors with a p-value < 0.20 in univariable models were retained for investigation in a multivariable model [64].

Variables retained from the univariable models were included together in a maximal multivariable negative binomial regression model. The interaction of any retained climate variables was assessed in the multivariable model. Manual backwards elimination was used to remove variables from the maximal model that did not have a significant association with risk in the multivariable model based on a significance threshold of p < 0.05 [64]. Potential confounding was investigated at each step by comparing model coefficients before and after the removal of each variable; a change of more than 20% indicated that a variable should be retained as a confounder regardless of its statistical significance in the model [64]. The change in Akaike Information Criterion (AIC) between the maximal and reduced model was assessed. Choropleth maps with Jenks natural breaks were used to visualize ZCTA-level CI of LACV-ND from 2015–2020 along with the distribution of significant predictors retained in the final model.

**Local model.** A geographically weighted (i.e., local) regression model was fitted to explore spatial variation in the relationships between predictors and LACV-ND CI in the final global model. Geographically weighted regression produces location-specific model coefficients based on estimated bandwidths, allowing the user to explore spatial variations in the relationship between predictors and the outcome. A geographically weighted negative binomial regression model was fitted using a SAS macro in SAS software version 9.4 with the same outcome, offset, and predictors from the final global model [65–67]. A global (i.e., spatially constant) overdispersion parameter was specified to model overdispersion and a biquadratic kernel weighting function with adaptive bandwidth estimation was used to allow the size of the bandwidth to vary based on the density of observations [68]. The second-order Akaike Information Criterion (AICc) was used to identify the optimal bandwidth and to compare goodness-of-fit between the global and local models. A randomization non-stationarity test with 999 replications was used to test the stationarity of the local model coefficients (i.e., if the model coefficients changed significantly across the study area) [68], and the local coefficients of non-stationary variables were visualized as choropleth maps. To determine whether the predictors maintained a significant association with risk throughout the entire area, choropleth maps of only the locally significant coefficients were created.

## Results

### Reported cases

A total of 521 cases of LACV-ND were reported among people residing in the states of Tennessee and North Carolina from 2003–2020, of which 486 (92%) were reported within the study area of eastern TN (192 cases) and western NC (294 cases) (Table 1). Within the study area,

**Table 1. Number of reported LACV-ND cases by state and age from 2003–2020.**

| Age Group (Years) | Total cases (%) | Eastern Tennessee (%) | Western North Carolina (%) |
|---|---|---|---|
| 0 to 9 | 292 (60%) | 129 (67%) | 163 (55%) |
| 10 to 19 | 129 (27%) | 57 (30%) | 72 (25%) |
| 20 to 39 | 23 (5%) | 3 (1.5%) | 20 (7%) |
| 40 to 59 | 16 (3%) | 1 (0.5%) | 15 (5%) |
| 60 to 79 | 21 (4%) | 2 (1%) | 18 (6%) |
| 80 and older | 5 (1%) | 0 (0%) | 5 (2%) |
| **Total** | 486 | 192 | 294 |

275 reported cases were males (58%) and 198 (42%) were females, while an additional 13 case reports did not include sex. Most cases (421 cases, 87% of total) occurred in people aged 19 years and younger. There was a notable discrepancy between the relative proportion of LACV-ND reported in adults between Tennessee and North Carolina. In eastern TN, only 3% of cases were reported in people older than 19 years, but in western North Carolina, 20% of cases were reported in people older than 20 years.

An annual mean of 27 LACV-ND cases were reported among all ages, but trends in reported cases did not always correlate between eastern TN and western NC (Fig 2A). Overall, most cases occurred between June and October, and more cases were reported in August than any other month. Of note, a higher percentage of cases occurred in the late spring and early

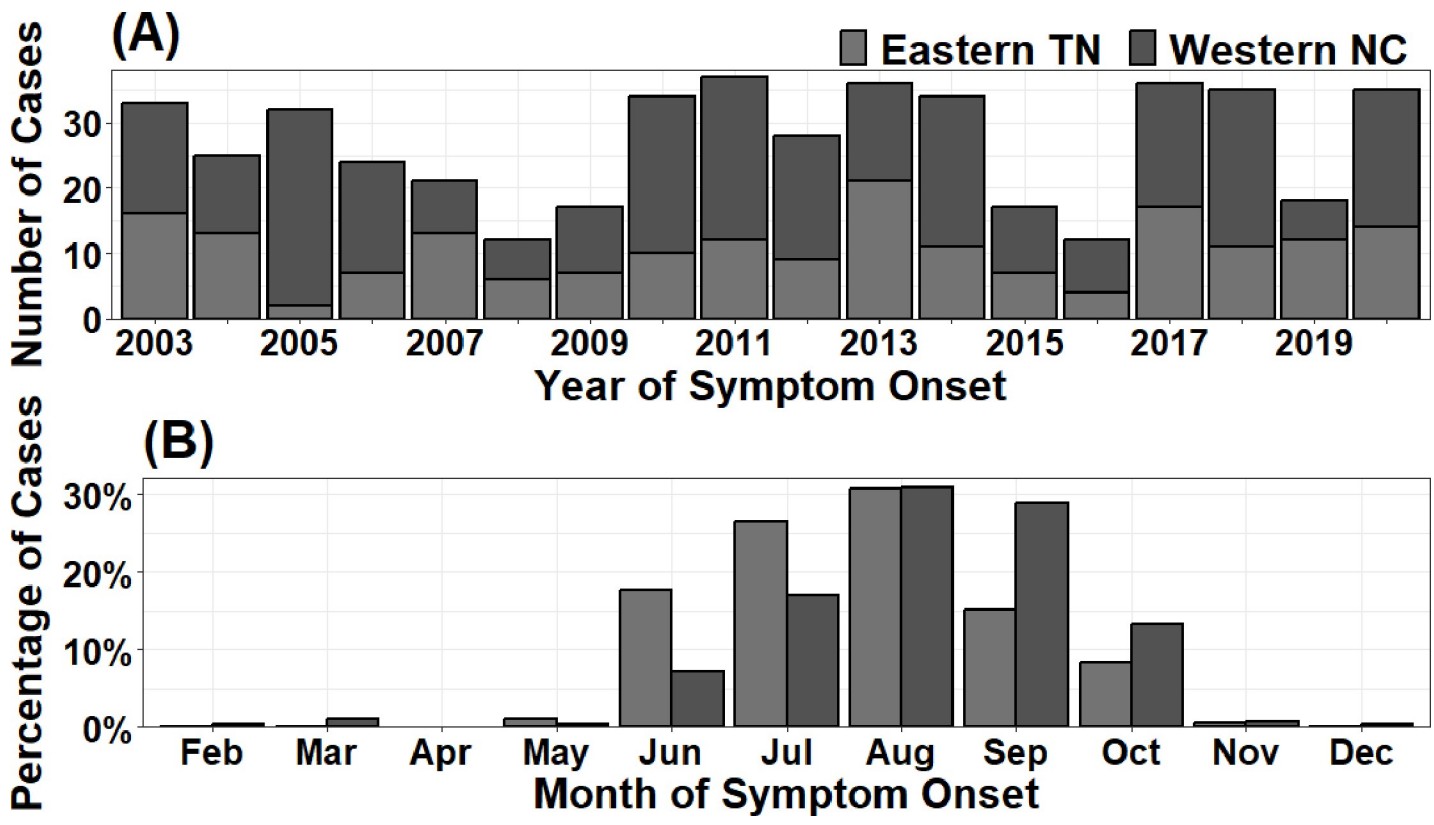

**Fig 2.** Annual number of reported probable and confirmed cases of La Crosse virus neuroinvasive disease (LACV-ND) (A) and the monthly percentage of total cases (B) in eastern Tennessee (TN) and western North Carolina (NC) from 2003 to 2020.

summer in eastern TN than in western NC, where more cases occurred in the late summer and early Autumn (Fig 2B).

## Objective 1: Cluster investigation

The overall CI of LACV-ND among the under-19 population was 2.57 cases per 10,000 persons from 2003–2011 and 2.72 cases per 10,000 persons from 2012–2020. The ZCTA-level CI during that period ranged from 0 to 209.9 cases per 10,000 persons from 2003–2011 and 0 to 222.2 cases per 10,000 persons from 2012–2020 (Fig 3). During both periods, most ZCTAs had zero cases (77% of ZCTAs from 2003–2011, 76% of ZCTAs from 2012–2020). Choropleth maps revealed a consistent risk distribution between the periods, with the highest risks occurring in the north-central region of eastern TN and the southwestern region of western NC throughout the entire study period, with a slight southward expansion of high-risk ZCTAs in eastern TN during 2012–2020 (Fig 3).

There were three significant purely spatial clusters of LACV-ND risk during the period 2003–2011, including two clusters in southwestern NC and one cluster in the north-central region of eastern TN (Fig 4A). The CI within the clusters ranged from 14.2–26.7 per 10,000 persons, and relative risk within the clusters ranged from 5.5 to 10.4 times higher than in the study area overall (Table 2). These clusters accounted for 56% of all cases reported in the under-18 population during this period.

There were four significant purely spatial clusters of LACV-ND risk in the under-19 population during the period 2012–2020, including two clusters in southwestern NC and two clusters in the north-central region of eastern TN (Fig 4B). Three of the clusters were geographically similar to the ones identified from 2003–2011, but an additional cluster in TN

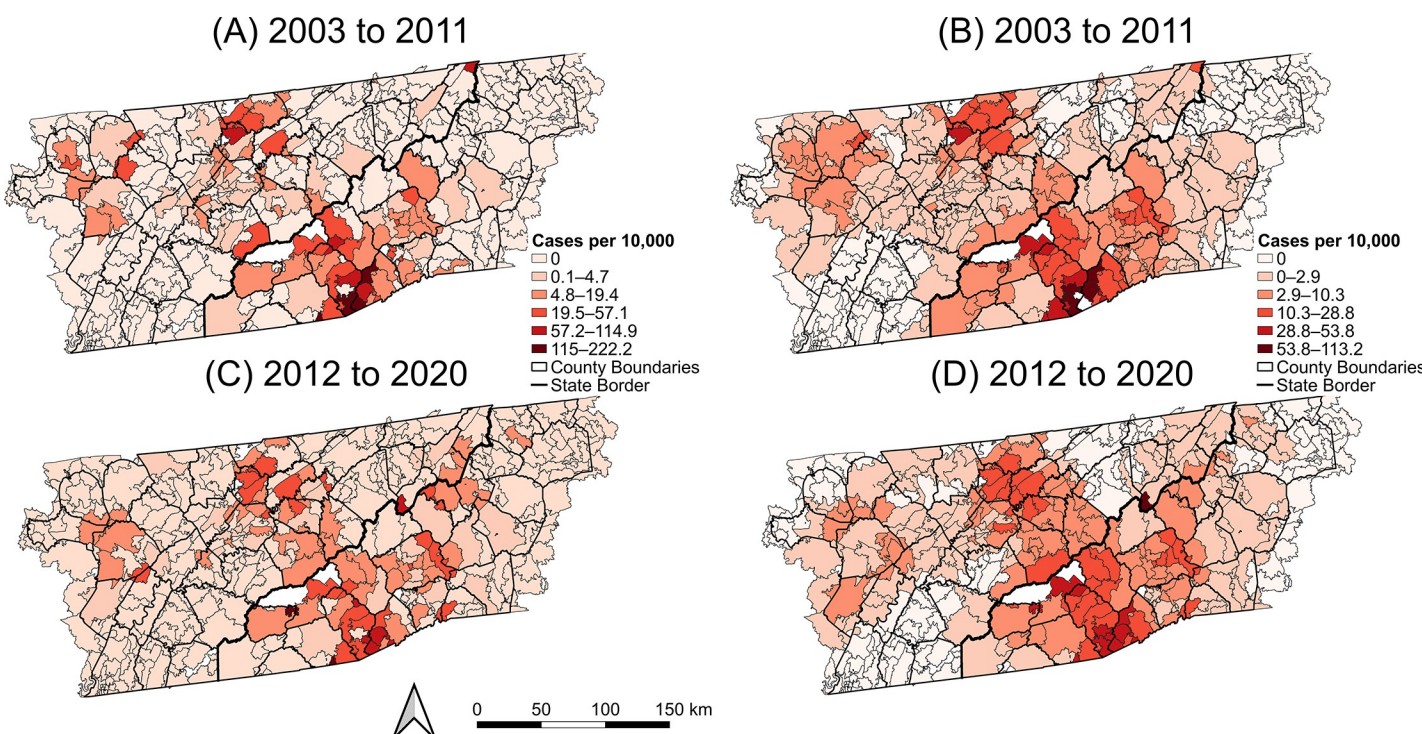

**Fig 3.** Raw (left) and smoothed (right) cumulative incidence of La Crosse virus neuroinvasive disease for the population 18 years and younger from 2003–2011 (A, B) and 2012–2020 (C, D). Base layers were derived from the 2010 United States Census Bureau TIGER/Line files [38].

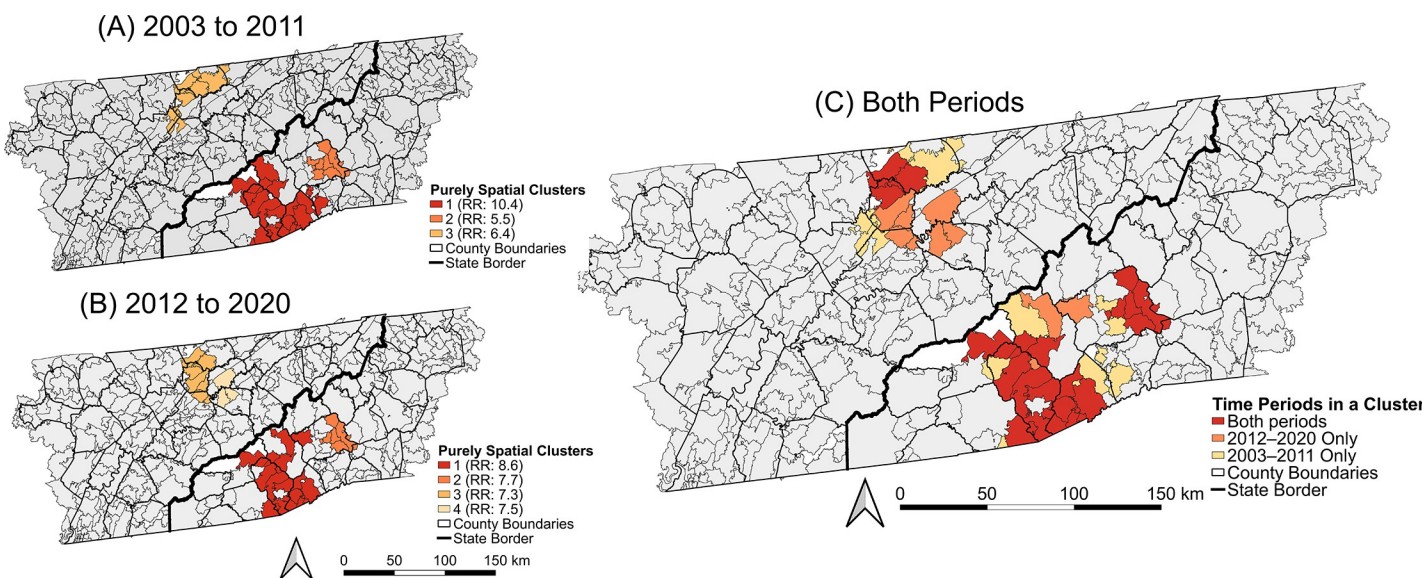

**Fig 4.** Purely spatial high-risk clusters and relative risk (RR) of La Crosse virus neuroinvasive disease in the population and years and younger from 2003–2011 (A) and 2012–2020 (B), and the time periods that each area was in a high-risk cluster (C). Base layers were derived from the 2010 United States Census Bureau TIGER/Line files [38].

reflected the expansion in CI in the period 2012–2020 (Fig 4B). The CI in the clusters ranged from 19.8–24.1 per 10,000 persons, and relative risk within the clusters 7.3 to 8.9 times higher than the study area overall. The clusters accounted for 50% of all cases reported in the under-19 population during this period.

**Table 2. Purely spatial and space-time clusters of La Crosse virus disease from 2003–2020.**

| Cluster Type | Period | Cluster (State) | Number of ZCTAs[1] | Cases (Expected) | Total population under 18 years | CI[2] per 10,000 population under 18 years | RR[3] | p-value |
|---|---|---|---|---|---|---|---|---|
| Purely Spatial (High Risk) | 2003–2011 | 1 (NC) | 19 | 62 (6) | 23,240 | 26.7 | 10.4 | < 0.001 |
| | | 2 (NC) | 8 | 30 (5.5) | 21,146 | 14.2 | 5.5 | < 0.001 |
| | | 3 (TN) | 8 | 19 (3.0) | 11,514 | 16.5 | 6.4 | < 0.001 |
| | 2012–2020 | 1 (NC) | 16 | 51 (5.8) | 21,151 | 24.1 | 8.9 | < 0.001 |
| | | 2 (NC) | 6 | 26 (3.4) | 12,426 | 21.0 | 7.7 | < 0.001 |
| | | 3 (TN) | 7 | 20 (2.7) | 10,088 | 19.8 | 7.3 | < 0.001 |
| | | 4 (TN) | 3 | 13 (1.7) | 6,355 | 20.5 | 7.5 | 0.002 |
| Space-Time (High Risk) | 2003–2018 | 1 (NC) | 26 | 110 (16.9) | 36,703 | 30.0 (1.9/year) | 8.5 | < 0.001 |
| | 2003–2017 | 2 (NC) | 6 | 40 (5.2) | 12,004 | 33.3 (2.2/year) | 8.4 | < 0.001 |
| | 2007–2020 | 3 (TN) | 13 | 44 (7.8) | 19,308 | 22.8 (1.6/year) | 6.2 | < 0.001 |
| Space-Time (Low Risk) | 2003–2018 | 1 (TN/NC) | 66 | 1 (66.8) | 143,570 | 0.07 (0.004/year) | 0.01 | < 0.001 |
| | 2003–2018 | 2 (TN/NC) | 94 | 9 (73.6) | 160,125 | 0.56 (0.04/year) | 0.1 | < 0.001 |
| | 2003–2018 | 3 (TN) | 25 | 0 (26.1) | 56,738 | 0 | 0 | < 0.001 |

[1]ZIP Code Tabulation Area

[2]CI: Cumulative Incidence

[3]Relative Risk

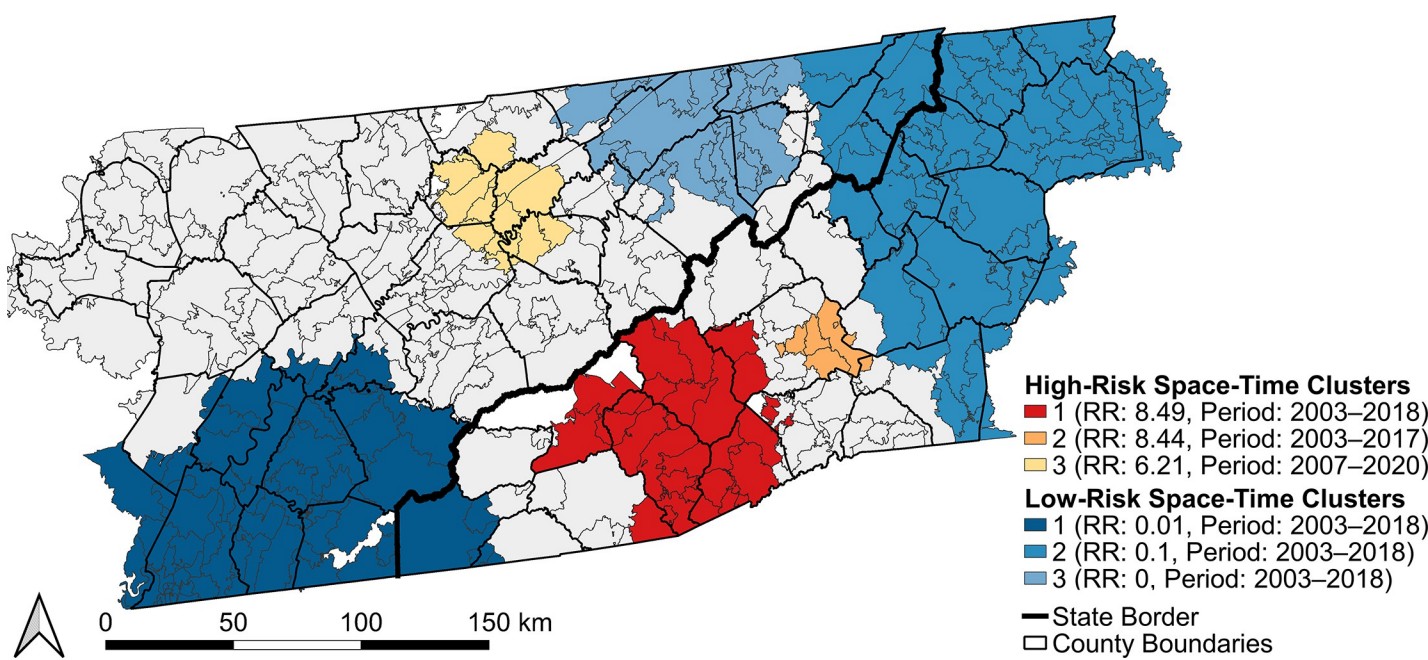

**Fig 5.** Statistically significant high-risk (red) and low-risk (blue) space-time clusters of La Crosse virus neuroinvasive disease in the population 18 years and younger from 2003–2020 with relative risk (RR), temporal period of clustering, and cluster ranks in order of maximum likelihood. Base layers were derived from the 2010 United States Census Bureau TIGER/Line files [38].

There was considerable overlap in the distribution of high-risk clusters between the two time periods. Of the 45 ZCTAs that were in a cluster during at least one period, 22 (49%) were in clusters during both periods. There was a slight southern expansion in high-risk clustering in eastern TN between the two periods, with a high-risk cluster in Grainger and Jefferson counties during 2012–2020 that was not present during 2003–2011 (Fig 4C). The ZCTAs included in the southwestern NC clusters changed slightly, but the overall geographic distribution of the clusters was similar between the two periods. In total, 53% of all pediatric

**Table 3. Maximal global negative binomial model of pediatric LACV-ND risk at the ZCTA level from 2015 to 2020.**

| Variable | Coefficient (95% Confidence Interval[1]) | p-value |
|---|---|---|
| Mean precipitation in August (mm) | 0.4 (0.2, 0.5) | < **0.001** |
| Mean temperature in August (˚C) | 1.9 (1.0, 2.9) | < **0.001** |
| Precipitation*Temperature | -0.02 (-0.02, -0.01) | < **0.001** |
| Population Density $Km^2$ | 0.001 (-0.001, 0.003) | 0.3 |
| Percent Vacant Housing | -0.001 (-0.03, 0.02) | 0.9 |
| Percent Housing Built Before 1969 | 0.003 (-0.02, 0.02) | 0.8 |
| Percent Forest in 2019 | 0.01 (-0.009, 0.04) | 0.2 |
| Percent Change from Forest to Developed Land from 2001–2019 | -0.05 (-0.2, 0.1) | 0.5 |

**Table 4.  Results of final global negative binomial regression model of LACV-ND risk from 2015–2020.**

| Variable | Coefficient (95% Confidence Interval[1]) | p-value |
|---|---|---|
| Mean precipitation in August (mm) | 0.4 (0.2, 0.5) | < **0.001** |
| Mean temperature in August (˚C) | 1.7 (0.9, 2.5) | < **0.001** |
| Precipitation*Temperature | -0.02 (-0.5, 0.1) | < **0.001** |

LACV-ND cases in the study area were reported from people residing within the spatial clusters identified here.

There were three significant and persistent high-risk space-time clusters from 2003–2020: two in southwestern NC and one in the north-central region of eastern TN (Fig 5). Risk in the clusters ranged from 6.2- to 8.5-times higher than outside the clusters (Table 2). The high-risk space-time clusters overlapped with the purely spatial clusters (Figs 4 and 5). There were also three significant and persistent low-risk space-time clusters from 2003–2020: one in the southwestern region of the study area, one in the eastern region, and another in the northeast region of eastern TN (Fig 5). Risk in the low-risk clusters ranged from 90% lower to 100% lower than the study area overall (Table 2).

## Objective 2: Predictor investigation

**Global model.**   Because of their strong correlations with other variables, elevation (correlated with mean temperature in August, r = -0.96), percentage of developed land (correlated

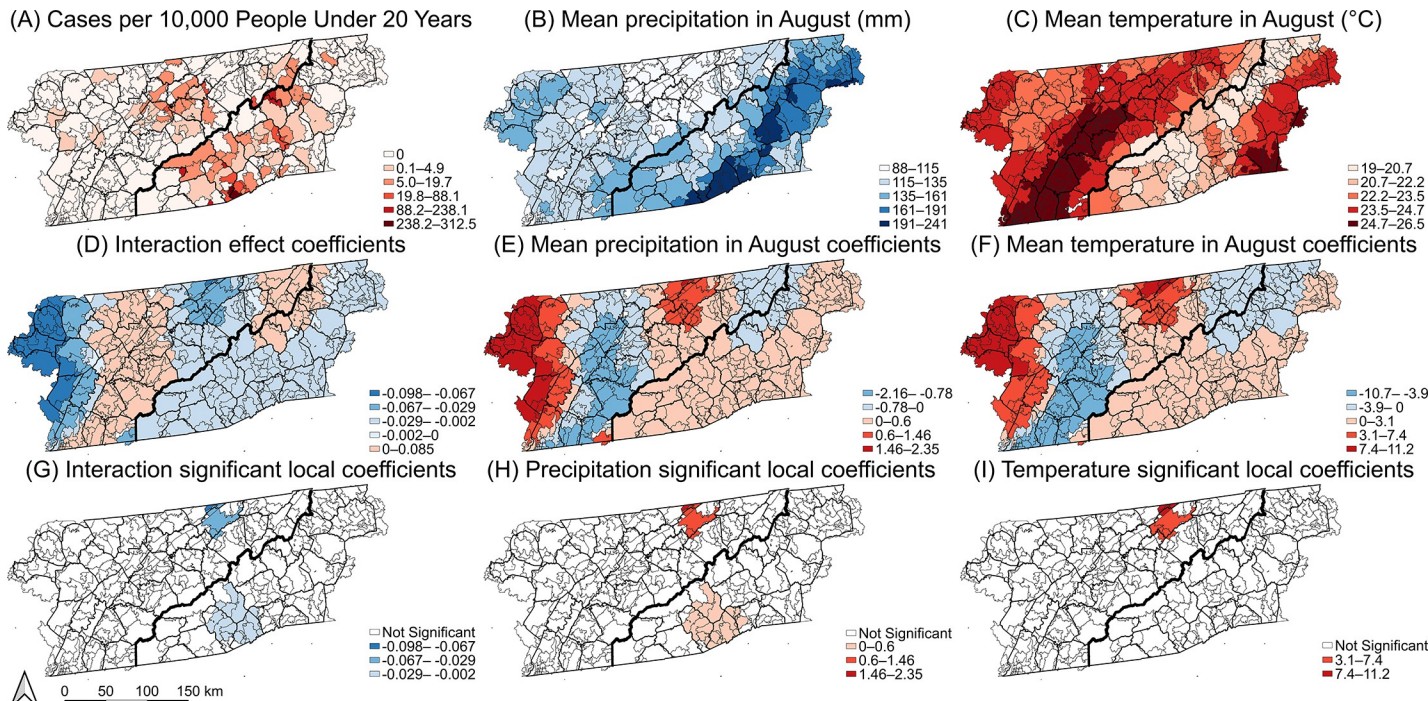

**Fig 6.**  Geographic distributions of ZCTA-level LACV-ND cumulative incidence risk in the population 19 years and younger (A), the distribution of significant predictors (B,C), the distribution of local coefficients (D–F), and the distribution of locally significant coefficients (G–I). Base layers were derived from the 2020 United States Census Bureau TIGER/Line files [53].

with population density, r = 0.89, and percentage of forested land, r = -0.77), and dew point (correlated with temperature, r = 0.94) were not considered as potential predictors. Descriptive statistics for the remaining variables are presented in S3 Table.

Seven potential predictors were associated with LACV-ND CI in univariable models (S4 Table). After fitting those predictors in a maximal multivariable model, only mean precipitation in August, mean temperature in August, and their interaction maintained a significant association with pediatric LACV-ND incidence (Table 3). Manual backwards elimination identified a model that only included the climate variables as significant predictors. Risk was higher in areas with higher mean precipitation or with higher mean temperature in August, but the two variables had an antagonistic interaction effect (Table 4). The AIC of the reduced model was lower than the maximal model ($\Delta$AIC = 7.7)

**Local model.**   The local model containing mean precipitation in August, mean temperature in August, and their interaction in the years 2015–2020, had a substantially better fit than the global model ($\Delta$AICc = 32). There was significant evidence of non-stationarity in the coefficients of mean precipitation in August ($p < 0.001$), mean temperature in August ($p < 0.001$), and their interaction ($p < 0.001$), and a visual assessment of the local coefficients further demonstrated that the relationships between each predictor and LACV-ND risk varied significantly across the study area (Fig 6). Only a small proportion of ZCTAs had locally significant relationships between the predictors and LACV-ND risk. Temperature and precipitation were positively associated with risk in a small region of eastern TN, but there were no cases in that area. A subset of high-risk ZCTAs in southwestern NC had significant positive local associations with precipitation, but not temperature (Fig 6). The interaction effect was locally significant in all areas where precipitation or temperature were significant.

## Discussion

The results of this study demonstrate that ZCTA-level LACV-ND risk clustering is spatially and temporally persistent in eastern TN and western NC and provides first evidence of spatially heterogenous associations between climate and LACV-ND risk. Because risk remains spatially consistent for extended periods, reactive public health interventions targeted at areas with documented LACV transmission is a viable approach to reducing LACV-ND risk. Climate variables may also be useful for predicting areas with high risk of LACV transmission, but the results of this exploratory indicate that the strength of the relationship between climate and disease risk is heterogenous in this study area. Based on these findings, targeted interventions in persistent high-risk areas of eastern TN and western NC along with further research on the role of climate in the ecology and epidemiology of LACV transmission are warranted.

The focal persistence of LACV-ND has been demonstrated in previous studies [15]. Early surveillance efforts in La Crosse, Wisconsin identified persistent LACV exposure in a single residential community, where targeted vector control substantially reduced exposure rates [29]. In Illinois from 1966–1995, approximately 40% of all cases consistently occurred around a single town, and more recently, the identification of noncoincident LACV-ND cases at individual residences in western North Carolina provided evidence that LACV-ND risk is persistent at the household level [23,24]. The findings of the present study corroborate existing evidence that LACV transmission risk remains focally persistent for extended periods, with more than half of all pediatric LACV-ND cases occurring in a few persistent high-risk foci. Where there is evidence of LACV transmission to humans, public health officials should expect that the virus will continue to be transmitted in that area in the absence of interventions.

Transovarial transmission of LACV by the principal vector, *Aedes triseriatus*, is a potential mechanism for the focal persistence of LACV-ND. A proportion of eggs laid by LACV-

infected *Ae. triseriatus* females are often infected with LACV, and this vertical transmission to diapausing eggs is the mechanism by which LACV persists throughout the winter when adult mosquitoes are inactive [69]. However, the reported rates of vertical and horizontal zoonotic LACV transmission are too low to account for long-term maintenance of viral transmission cycles, leaving the question of focal persistence open ended [2]. The leading hypothesis is that some *Ae. triseriatus* achieve stabilized infections, which would reduce the inefficiencies of the vertical transmission pathway and potentially allow a small number of stably infected mosquitoes to facilitate the natural persistence of LACV within focal areas [70].

Climate is known to be an important driver of mosquito-borne diseases, but the role of climate in LACV transmission is not understood [16,60,71,72]. This study provides evidence that average precipitation and temperature in the month of August have positive, but antagonistic, relationships with the geographic risk of LACV-ND risk. However, local regression models revealed significant spatial heterogeneity in the strength and significance of those climatic relationships, revealing a lack of locally significant climatic relationships with risk in most of the study area. These findings are possibly related to the distinctly different climates present in western NC and eastern TN. Southwestern NC shares environmental similarities with the GSMNP, including high humidity and precipitation consistent with temperate rainforest climates. In contrast, most of eastern TN is at a lower elevation with higher temperatures and less precipitation than southwestern NC, which likely results in differences in vector phenologies and LACV seasonality between the states [73]. To that point, a higher percentage of cases in eastern TN occurred in the late spring and early summer than in western NC, which may reflect faster seasonal development of mosquitoes in the warmer climate of eastern TN than the cooler temperatures and higher elevation of western NC.

Notably, this investigation used climate values from the month of August, the peak month of reported LACV-ND in the region, to compare late summer temperature and precipitation throughout the study area. Climate in earlier months likely plays an important role in subsequent LACV transmission by effecting the population and survival sizes of vectors and wildlife hosts [74]. Further research that investigates climate-driven LACV maintenance and transmission throughout the study area would provide a better understanding of the role that climate plays in LACV-ND risk, and long-term LACV surveillance of vectors and wildlife hosts would be especially useful for developing time series models to assess the utility of climatic predictors from different points of the year for forecasting local transmission risk.

Previous studies found that LACV-ND risk was higher in areas with lower socioeconomic metrics [11,27], but in the present study, socioeconomic variables were not strongly associated with disease risk. This finding is likely due to differences in study area and unit of geographic inference compared to previous studies. Day, Odoi, et al. (2023) [11] investigated county-level data for the entire eastern US, and Haddow et al. (2011) [27] investigated predictors for all census tracts in West Virginia. In both studies, LACV-ND risk was highly clustered in focal regions of the study area where socioeconomic metrics were relatively low, while the study area included larger regions that had minimal reported disease. The study area in this investigation had a smaller geographic range than previous studies, and primarily included counties with relatively high LACV-ND incidence risks and low socioeconomic metrics relative to the eastern US overall [11]. By focusing on a smaller geographic area, this study revealed that within the high-risk region of eastern TN and western NC, risk is not closely associated with population-level socioeconomic status. However, there remains the potential for individual-level associations that cannot be inferred with a population-level study. For example, there are several known household-level risk factors for LACV transmission, such as the presence of artificial containers, proximity to tree holes, and *Aedes* mosquito abundance, which may each be related to socioeconomic status at the household level [29–33].

In this study, a global negative binomial regression model was used to identify significant predictors, while geographically weighted regression was used to fit a local model for exploration spatial variation in model coefficients. Geographically weighted regression is considered an exploratory approach for identifying variation in model coefficients, rather than a statistical framework for inferring the causal relationships between predictor variables and an outcome [75]. Consequently, the results of the predictor investigation are best understood as an exploration of relationships between hypothesized predictors and LACV-ND risk. The findings demonstrate that although climate values had the strongest associations with risk, a single set of model coefficients is not sufficient for representing the relationships between climate variables and risk is this study area.

Potential case reporting bias is an important limitation of this study. The underreporting of LACV infections is a known problem that causes the burden of LACV-ND in the US to go under-measured [5,76], and it is possible that within the low-risk areas identified here, clinicians are simply less likely to test for, diagnose, or report LACV-ND than clinicians in high-risk areas. Enhanced surveillance throughout the study area would be useful for determining if LACV infection risk is truly minimal in the low-risk areas identified here. Serosurveys of humans and wildlife, in particular, are effective for determining whether LACV is circulating in an environment where clinical reporting bias is expected [33,77,78]. Reporting bias may also have occurred due to changes in the case definition of LACV-ND during the study period. For example, in 2011, the case definition changed to require a measured fever $> 38\degree C$, which may have reduced the number of reported LACV-ND cases relative to the current case definition, which does not include a measured fever [36,79]. The assumption that LACV exposure occurred at home residences is a related limitation that may have biased the cluster analyses and predictor investigation. Although LACV-ND is often associated with LACV-infected mosquitoes at home residences [33], it is likely that many cases in this dataset arose from pathogen exposure in other areas. For example, there is evidence of high LACV exposure rates in US National Park Service employees working at the GSMNP, implying that visitors are also at risk of exposure in the park [80]. Any LACV-ND that developed from exposure within the national park would be inaccurately associated with home residences, even if the residence is outside the endemic range of LACV-ND [81].

## Conclusions

There are persistent clusters of pediatric LACV-ND within the regional disease hotspot of eastern Tennessee and western North Carolina. Approximately half of all cases reported from 2003 to 2020 occurred within a few persistent high-risk clusters that were adjacent to large regions with minimal reported disease. Because focal risk clusters persist for extended periods, retroactive surveillance is useful for guiding targeted vector control and community education programs to reduce continued transmission in high-risk areas. The distribution of ZCTA-level risk from 2015–2020 was positively associated with mean temperature and precipitation in August, but the interaction between those two variables was antagonistic, and a local regression model revealed that the climatic relationships were risk were not locally significant throughout most of the study area. Further research on the role of climate in LACV risk is warranted to support the potential development of predictive models to support proactive interventions that do not rely solely on retroactive surveillance.

## Supporting information

**S1 Table. Data sources, temporal resolution, spatial resolution, and usage.**
(DOCX)

**S2 Table. Potential predictor variables, rationale for inclusion, and associated references.** (DOCX)

**S3 Table. Descriptive statistics of potential predictor variables.** (DOCX)

**S4 Table. Results of univariable global negative binomial regression models (p-values < 0.20 in bold).** (DOCX)

## Acknowledgments

Sandy Lindsay and Allyson Graves from the University of Tennessee, Knoxville were invaluable in negotiating and executing data use agreements. Dr. Jennifer Lord, also from the University of Tennessee, Knoxville, provided valuable assistance with implementing the SAS macro for the negative binomial geographically weighted regression model.

## Author Contributions

**Conceptualization:** Corey A. Day, Agricola O. Odoi, Abelardo Moncayo, Michael S. Doyle, Carl J. Williams, Brian D. Byrd, Rebecca T. Trout Fryxell.

**Data curation:** Corey A. Day, Abelardo Moncayo, Michael S. Doyle.

**Formal analysis:** Corey A. Day.

**Investigation:** Corey A. Day.

**Methodology:** Corey A. Day, Agricola O. Odoi.

**Supervision:** Agricola O. Odoi, Rebecca T. Trout Fryxell.

**Validation:** Corey A. Day.

**Visualization:** Corey A. Day.

**Writing – original draft:** Corey A. Day.

**Writing – review & editing:** Corey A. Day, Agricola O. Odoi, Abelardo Moncayo, Michael S. Doyle, Carl J. Williams, Brian D. Byrd, Rebecca T. Trout Fryxell.

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
