## [Decision Letter · Decision Letter 0]

11 Jan 2024

Dear Dr. Day,

Thank you very much for submitting your manuscript "Persistent spatial clustering and predictors of pediatric La Crosse virus neuroinvasive disease risk in eastern Tennessee and western North Carolina, 2003–2020" for consideration at PLOS Neglected Tropical Diseases. As with all papers reviewed by the journal, your manuscript was reviewed by members of the editorial board and by several independent reviewers. In light of the reviews (below this email), we would like to invite the resubmission of a significantly-revised version that takes into account the reviewers' comments. 

All reviewers provided strong support for the manuscript, yet suggested small but important clarifications should be added. Should the authors wish to submit a revision, they should address these clarifications to the methods, and include discussion of the additional limitations as suggested by the reviewers. Reviewer 3 brought up an excellent point concerning humidity, which the authors absolutely must address.

We cannot make any decision about publication until we have seen the revised manuscript and your response to the reviewers' comments. Your revised manuscript is also likely to be sent to reviewers for further evaluation.

Sincerely,

Jeremy V. Camp, Ph.D.

Academic Editor

Michael Holbrook

Section Editor

All reviewers provided strong support for the manuscript, yet suggested small but important clarifications should be added. Should the authors wish to submit a revision, they should address these clarifications to the methods, and include discussion of the additional limitations as suggested by the reviewers. Reviewer 3 brought up an excellent point concerning humidity, which the authors absolutely must address.

Reviewer's Responses to Questions

**Key Review Criteria Required for Acceptance?**

**Methods**

-Are the objectives of the study clearly articulated with a clear testable hypothesis stated?

-Is the study design appropriate to address the stated objectives?

-Is the population clearly described and appropriate for the hypothesis being tested?

-Is the sample size sufficient to ensure adequate power to address the hypothesis being tested?

-Were correct statistical analysis used to support conclusions?

-Are there concerns about ethical or regulatory requirements being met?

Reviewer #1: 1. This study used data from 2023-2022. Why climate data used only 2015-2020?

2. Why focus on 18 years and under? Why page 6 line 142 change to 19 years and younger?

Reviewer #2: Yes. The study design is appropriate. I recommend outlining in detail some limitations of Geographically Weighted Regression in the limitation section of the manuscript.

Reviewer #3: 1-Land cover data methods section- It is unclear why land change from 2001 to 2019 was used, when the case data was from 2003-2020. Please correct accordingly or clarify rationale for temporal variation.

2-Climate data methods section- It is unclear why climate data for five years was used when case data was for 17 years. Again, please correct accordingly or clarify rationale for temporal variation. Note: PRISM data goes back decades allowing for overlapping datasets (climate and case).

3-Climate data methods section- Why is humidity not included in this data set? Humidity is arguably just as important in mosquito biology as temperature and precipitation. Please re-rerun the analyses with this factor, or provide rationale as to why it was not included. https://www.ncbi.nlm.nih.gov/pmc/articles/PMC10299817/

4-Land cover, climate and elevation predictors methods section- The rationale for using a one year snapshot of land cover classification types is unclear. Why would forest land cover in 2019 have any prediction relationship with cases in 2003? If you are using cumulative incidence as a measure, this should be clarified.

**Results**

-Does the analysis presented match the analysis plan?

-Are the results clearly and completely presented?

-Are the figures (Tables, Images) of sufficient quality for clarity?

Reviewer #1: 1. In the study area was 486 cases but male 275 + female 198 = 473. What happen to other 13 cases? Page 11 line 283-285

2. In the method section focus on 18 years and younger, why in the result become 15 years and younger? What is the criteria to classified age groups in table 1?

Reviewer #2: Yes.

Reviewer #3: (No Response)

**Conclusions**

-Are the conclusions supported by the data presented?

-Are the limitations of analysis clearly described?

-Do the authors discuss how these data can be helpful to advance our understanding of the topic under study?

-Is public health relevance addressed?

Reviewer #1: -

Reviewer #2: Yes. I recommend outlining limitations of Geographically Weighted Regression in the limitation section

Reviewer #3: (No Response)

**Editorial and Data Presentation Modifications?**

Reviewer #1: -

Reviewer #2: The study was well organized and well executed, and the results make a nice contribution to ongoing efforts to understand patterns of LaCrosse encephalitis virus risk and correlations with socioeconomic and environmental data. The models and results are appropriate for the analyses, with one caveat regarding geographically weighted regression, which could be outlined in the limitations section of the manuscript. 

Specifically, there has been some debate in the literature about whether GWR is more of a descriptive modeling approach and what this means when inferring model results. I recommend outlining in the study limitations some of the limitations of GWR and pointing out that additional modeling frameworks are also available to analyze spatial areal data, including conditional autoregressive (CAR) models and simultaneous autoregressive models (SAR) models. 

Other than this caveat, I have only minor suggestions for clarity to help improve readability of the manuscript:

Under the software heading, spatial scan statistics are not mentioned and only descriptive statistics are associated with R.

Under the heading Global Model Fitting, it would be helpful to the reader to clarify here and indicate that this is the first part of the geographically weighted regression (GWR). Please state that a generalized linear model or a generalized linear mixed effects model (negative binomial) was used.

Reviewer #3: Minor edits:

1-The introduction could be strengthened with more descriptive facts. The introduction comes across very vague and fails to 'sell' the reader on the importance and value of this article. For example, you could state "LACV has grave economic consequences--a single case can accrue up to $3 million in projected costs--a particularly concerning impact considering socioeconomically disadvantaged populations are at greatest risk for this disease." Ref: https://pubmed.ncbi.nlm.nih.gov/14695088/ and https://pubmed.ncbi.nlm.nih.gov/21980533/

2- IRB approval for TN is noted, but not NC. If NC health department did not require internal IRB approval, this should be stated to clarify ethical approval processes.

3-Lines 153-155. Please describe the extent of 'developed land' used. National remote sensing standards use five 'developed' land classifications (see link). Did you consider any change in development category apart of 'percent change in developed land' or what metric did you use? https://www.mrlc.gov/data/legends/national-land-cover-database-class-legend-and-description. 

4-Lines 239-240: Authors mention an important point and study limitation. Perhaps using underlying Ecoregion IV would be a better indicator of environment than 'forest'. Forest is defined as three different categories per national remote sensing standards (which forest type should be clarified), yet none of these get to the detail level of at-risk individual tree species ecology. https://www.mrlc.gov/data/legends/national-land-cover-database-class-legend-and-description

5-Results section: Overall this section reads very verbose. I would encourage the senior author(s) to refine and streamline this section.

6-Table 3 (line 351): Given the large number of tables in the paper, this one would make a good candidate for a supplemental table OR authors could combine tables 3 and 4.

7-Lines 362-363: I am not surprised you only found a small number of zipcodes with statistically significant associations. This would likely change if you used the same years for both datasets--see prior comments on non-matching temporal datasets.

8-Limitation: Authors fail to mention that Aedes japonicus is another important LACV vector with a differing ecology than Aedes triseratus. The ecology for this secondary vector is not evaluated by the current paper's methods and future studies should evaluate this additional vector. 

9-Figure 5: The legend is incorrect. Your low-risk clusters all list '2003-2018'.

**Summary and General Comments**

Reviewer #1: The authors present an analysis during 2003 to 2022 of data on La Crosse virus. Overall the paper is well written; however, there are a few points that need to be reassessed and strengthened.

Reviewer #2: The objective of this study is to analyze the spatiotemporal distribution of pediatric LaCrosse Encephalitis virus neuroinvasive disease between 2003 and 2020 in the Appalachian Mountains region of Tennessee and western North Carolina where consistent transmission occurs. The authors used multiple methods to investigate spatial clusters and space-time clusters, and they used a geographically weighted regression to investigate effects of covariates on observed patterns. All analyses were performed at the zip code tabulation area level. 

The study was well organized and well executed, and the results make a nice contribution to ongoing efforts to understand patterns of LaCrosse encephalitis virus risk and correlations with socioeconomic and environmental data. The models and results are appropriate for the analyses, with one caveat regarding geographically weighted regression, which could be outlined in the limitations section of the manuscript.

Reviewer #3: The authors perform spatial modelling to identify persistent areas of pediatric La Crosse encephalitis. While rare, this disease has important mortality and morbidity implications and leveraging geospatial techniques has promise to inform public health interventions. The authors are particularly applauded for including key state public health officials, who have the capacity to employ these models into their respective state's public health activities. 

Some core methodological issues inhibit the rigor and potential validity of the current analysis. These issues are all easily addressable, and pending their incorporation would make for an appropriate article that would be a welcomed addition to the scientific literature. Detailed major and minor comments are listed in prior reviewer sections.

PLOS authors have the option to publish the peer review history of their article (what does this mean?). If published, this will include your full peer review and any attached files.

Reviewer #1: No

Reviewer #2: No

Reviewer #3: No
---

## [Decision Letter · Decision Letter 1]

29 Mar 2024

Dear Dr. Day,

Thank you very much for submitting your manuscript "Persistent spatial clustering and predictors of pediatric La Crosse virus neuroinvasive disease risk in eastern Tennessee and western North Carolina, 2003–2020" for consideration at PLOS Neglected Tropical Diseases. As with all papers reviewed by the journal, your manuscript was reviewed by members of the editorial board and by several independent reviewers. The reviewers appreciated the attention to an important topic. Based on the reviews, we are likely to accept this manuscript for publication, providing that you modify the manuscript according to the review recommendations. 

Unfortunately, one of the reviewers of the initial submission was unavailable for a second round of peer review. We invited another reviewer (Reviewer 4), and they have provided some additional comments, while the two other first-round reviewers were satisfied with the revision.

Concerning the comments from Reviewer 4, the authors should respond to the question about "rflexscan" and the two main issues with Predictor Investigation (L275 and L304), as raised by Reviewer 4, and revise, if necessary. As as a major comment, Reviewer 4 also suggested to revise/reorganize the methods, and the authors should consider this, but this is not a requirement to be considered for acceptance. I agree with Reviewer 4 that the methods could be better organized (e.g., first item of the minor comments), but I consider this to be a stylistic/editorial request. However, please do improve the "under-citing" of methods, as suggested. Otherwise, the minor comments should be straightforward to address.

Sincerely,

Jeremy V. Camp, Ph.D.

Academic Editor

Michael Holbrook

Section Editor

Unfortunately, one of the reviewers of the initial submission was unavailable for a second round of peer review. We invited another reviewer (Reviewer 4), and they have provided some additional comments, while the two other reviewers were satisfied with the revision.

Concerning the comments from Reviewer 4, the authors should respond to the question about "rflexscan" and the two main issues with Predictor Investigation (L275 and L304), as raised by Reviewer 4, and revise, if necessary. As as a major comment, Reviewer 4 suggested to revise/reorganize the methods, and the authors should consider this, but this is not a requirement to be considered for acceptance. I agree with Reviewer 4 that the methods could be better organized (e.g., first item of the minor comments), but I consider this to be a stylistic/editorial request. However, please do improve the "under-citing" of methods, as suggested. Otherwise, the minor comments should be straightforward to address.

Reviewer's Responses to Questions

**Key Review Criteria Required for Acceptance?**

**Methods**

-Are the objectives of the study clearly articulated with a clear testable hypothesis stated?

-Is the study design appropriate to address the stated objectives?

-Is the population clearly described and appropriate for the hypothesis being tested?

-Is the sample size sufficient to ensure adequate power to address the hypothesis being tested?

-Were correct statistical analysis used to support conclusions?

-Are there concerns about ethical or regulatory requirements being met?

Reviewer #2: yes

Reviewer #3: (No Response)

Reviewer #4: (No Response)

**Results**

-Does the analysis presented match the analysis plan?

-Are the results clearly and completely presented?

-Are the figures (Tables, Images) of sufficient quality for clarity?

Reviewer #2: yes

Reviewer #3: (No Response)

Reviewer #4: (No Response)

**Conclusions**

-Are the conclusions supported by the data presented?

-Are the limitations of analysis clearly described?

-Do the authors discuss how these data can be helpful to advance our understanding of the topic under study?

-Is public health relevance addressed?

Reviewer #2: yes

Reviewer #3: (No Response)

Reviewer #4: (No Response)

**Editorial and Data Presentation Modifications?**

Reviewer #2: (No Response)

Reviewer #3: (No Response)

Reviewer #4: (No Response)

**Summary and General Comments**

Reviewer #2: The authors have addressed the comments and suggestions from the first review, and my recommendation is to accept the manuscript for publication.

Reviewer #3: (No Response)

Reviewer #4: This study presents a spatial analysis of pediatric La Crosse virus cases in an area with persistently high disease activity. Overall, the study is well implemented and will provide important guidance for public health planning and vector control activities in the study region. I also thought the authors provided a very nice description of the rationale behind user-specified parameters in the SaTScan analysis, which could serve as a model for other studies adopting this methodology. 

That said, I do have some major (and minor) concerns that the authors should address to strengthen the manuscript.

Major Comments

The Methods section would greatly benefit from considerable revision, mostly for cohesion, flow, and organization. Need to give definitions/explanations/rationale behind choices and data products before they are mentioned in text; as written, many sections felt "out of order" from the reader perspective. Concepts and terms are used in the manuscript, but introduced or expanded upon in later sections. For example, the description of switching rates to 19 years and under occurs much later in the methods (in the predictor investigation section), but is seemingly referenced in one of the cluster analyses. In general, I think many smaller sections in the methods could be combined, cohesively outlining the preparation of data inputs, then describing how each dataset was used for each statistical analysis. As written, the methods jump around, and it becomes difficult for the reader to follow which data are feeding into which analysis. Method choices are also under cited in many instances (i.e., provide citations for things like parameter choices, threshold choices, etc).

Curious as to why ‘rflexscan’ was used to implement purely spatial clusters, when SaTScan (which is used to implement the space-time scan statistic) also has this option. Was there a particular reason why Tango’s restricted flexibly shaped scan statistic was used? This is not a commonly used test, and I think it would be beneficial to briefly describe what this is doing, and why you chose to use it here. 

My biggest issues were with choices made in the implementation of the Predictor Investigation (specifically the two points below). This section should be revised, or could even be omitted given the strength of the other findings. 

L275 Climate variables were chosen for the month of August because of peak cases, but what about potential lags? I.e., might climate in the months preceding peaks of cases influence transmission via upstream influence on vector reproduction, or host populations? What is the mechanism of transmission that is being proposed by limiting to climate in August? (i.e., are you saying that climate in the month of transmission is saying something about the vector biology, transmission, human behavior that makes people susceptible to infectious bites?). 

L304 stepwise model selection procedures are notoriously problematic, often leading to locally (but not globally) optimal solutions. Suggest trying a multimodel selection framework, with an appropriate information criterion for ranking models, and seeing if this changes your final model. 

Minor Comments

Minor point for organization that echoes earlier comment about Methods…in the Methods section I recommend putting the Software paragraph after the description of statistical methods. As is, implementation is given for methods that haven’t been introduced yet, and it breaks up the case definition and case data sections (which you may be able to just combine under one heading anyway).

This was probably an error generated in the reviewer pdf, but Figure 1 was not included in the document.

L84 Don’t think I’ve seen the term “accessory vectors” used. Consider using “secondary” instead?

L95 Are these estimated total annual costs? Or lifetime management per case?

L103 Might be useful to mention that Ae. albopictus and Ae. japonicus are invasive here

L135 Consider using term other than hotspot here

L163 I think it would be helpful to show the difference in probable and confirmed case totals. If not in the main manuscript, a graph or table showing these totals by year/region would be worth adding to supplemental materials. 

West Nile virus is mentioned as another endemic arboviral disease in the area. Are cases frequently reported in the study region?

L181 Data were provided to home address. Need to describe aggregation of cases to zip codes. 

L192 Were there any changes in zip code boundaries during the study period?

L205 Jenks classification (not Jenk’s). Also provide a citation for use of Jenks classification for health data visualization (CDC has one, I believe)

L211 Specify if scan statistic was run on cases or rates (looks like rates, but need to specify)

L212 Was the spatial scan statistic run as two separate analyses, or is this saying that two different population denominators were used for respective time periods in the same analysis? Unclear as written.

L230 Why run separate analyses for high and low clusters? Looks like it’s because of different thresholds on cluster size for high versus low. Specify in text, and provide citations for these cutoffs. 

L245 check wording of this sentence

L268 “Data were”, check throughout

L269 Give versions for computer programs

L298 Need to provide citations for variable selection process and threshold values

L439 Are there any hypothesized mechanisms for why transmission is so focally persistent?

Fig 4, suggest not using blue to depict clusters with elevated risk, as this color is often used to depict cold spots in other common hotspot analyses (and used this way in space-time analysis).

PLOS authors have the option to publish the peer review history of their article (what does this mean?). If published, this will include your full peer review and any attached files.

Reviewer #2: No

Reviewer #3: No

Reviewer #4: No

Figure Files:

Data Requirements:

Reproducibility:

References

---

## [Editor Report · Decision Letter 2]

2 May 2024

Dear Dr. Day,

We are pleased to inform you that your manuscript 'Persistent spatial clustering and predictors of pediatric La Crosse virus neuroinvasive disease risk in eastern Tennessee and western North Carolina, 2003–2020' has been provisionally accepted for publication in PLOS Neglected Tropical Diseases.

Best regards,

Jeremy V. Camp, Ph.D.

Academic Editor

Michael Holbrook

Section Editor

---

## [Editor Report · Acceptance letter]

23 May 2024

Dear Dr. Day,

We are delighted to inform you that your manuscript, "Persistent spatial clustering and predictors of pediatric La Crosse virus neuroinvasive disease risk in eastern Tennessee and western North Carolina, 2003–2020," has been formally accepted for publication in PLOS Neglected Tropical Diseases.

Best regards,

Shaden Kamhawi

co-Editor-in-Chief

Paul Brindley

co-Editor-in-Chief
